# Longitudinal Relationship between Cognitive Function and Health-Related Quality of Life among Middle-Aged and Older Patients with Diabetes in China: Digital Usage Behavior Differences

**DOI:** 10.3390/ijerph191912400

**Published:** 2022-09-29

**Authors:** Zhihao Jia, Yan Gao, Liangyu Zhao, Suyue Han

**Affiliations:** School of Physical Education, Shandong University, Jinan 250061, China

**Keywords:** cognitive function, HRQoL, digital usage behavior, middle-aged and older people, diabetes

## Abstract

Background: Cognitive function and health-related quality of life (HRQoL) are important issues in diabetes care. According to the China Association for Aging, it is estimated that by 2030, the number of elderly people with dementia in China will reach 22 million. The World Health Organization reports that by 2044, the number of people with diabetes in China is expected to reach 175 million. Methods: Cohort analyses were conducted based on 854 diabetic patients aged ≥45 years from the third (2015) and fourth (2018) survey of the China Health and Retirement Longitudinal Study (CHARLS). Correlation analysis, repeated-measures variance analysis, and cross-lagged panel models were used to measure the difference in digital usage behavior in the established relationship. Results: The results show that the cognitive function of middle-aged and older diabetic patients is positively correlated with HRQoL. HRQoL at T1 could significantly predict cognitive function at T2 (PCS: B = 0.12, *p* < 0.01; MCS: B = 0.14, *p* < 0.01). This relationship is more associated with individual performance than digital usage behavior. Conclusions: Unidirectional associations may exist between cognitive function and HRQoL among middle-aged and older Chinese diabetes patients. In the future, doctors and nurses can recognize the lowering of self-perceived HRQoL of middle-aged and older diabetic patients, and thus draw more attention to their cognitive function, in turn strengthening the evaluation, detection, and intervention of their cognitive function.

## 1. Introduction

Cognitive impairment is one of the common complications in elderly patients [1]. For example, in China, the prevalence of mild cognitive impairments in the aging population (aged 60 and above) is 14.71% [2], and as age increases, its annual rate of progression to dementia is between 8% and 15% [2]. At the same time, cognitive impairment is also one of the common complications of diabetic patients. It is estimated that by 2045, about 170 million elderly people in China will be diabetic patients [3]. Therefore, the academic community has paid great attention to the cognitive function of the diabetic population, and found that the cognitive function of the diabetic population is closely related to the quality of life (QoL) of the elderly [4].

HRQoL is the perceived physical and mental health of an individual or group over time, including both physical component summary (PCS) and mental component summary (MCS) [5]. In contrast with the QoL, HRQoL pays special attention to the impact of disease and treatment process on the life of a person or a group [6]. Existing studies have found that changes in cognitive function are positively correlated with changes in HRQoL, and play a predictive role in future changes in HRQoL [7,8]. For example, cognitive decline was found to be a predictor of HRQoL decline in studies on multiple sclerosis patients, AIDS patients, and older women [4,9,10]. At the same time, some scholars have found that changes in HRQoL—whether it is the PCS or the MCS of HRQoL—can also predict cognitive changes in individuals in the future [11]. For example, Ezzati’s 2019 study demonstrated that changes in HRQoL preceded changes in cognition and predicted the occurrence of dementia [12]. Thus, cognitive function may be bi-directionally associated with HRQoL.

The bidirectional relationship between cognitive function and HRQoL may be more pronounced in terms of diabetic patients. This is because not only are people with diabetes 1–2 times more likely to develop cognitive risks than the general population [13,14], but this risk will increase over time [15]. At the same time, with the deepening of the research on HRQoL, the medical community generally believes that the core of diabetes management should include HRQoL maintenance in addition to prevention and delay of its complications [16]. In summary, this research aims to focus on middle-aged and older diabetic patients (over 45 years old) in China—not only because China has one of the largest numbers of diabetic patients in the world, but also because the age of the population affected with diabetes is showing a downward trend [3,17]. Thus, we propose the first hypothesis: in middle-aged and older Chinese patients with diabetes, the relationship between cognitive function and HRQoL may be bidirectional.

To our knowledge, previous studies on cognitive function and HRQoL have focused on unmodifiable clinical factors, such as age, gender, etc., with a lack of studies on modifiable factors [4]. Digital technologies such as the Internet and smartphones have attracted the attention of psychologists because of their portability, rapidity, and immediacy. In clinical medicine, digital usage behaviors are often used for managing and intervening with patients with diabetes and cognitive dysfunction [18]. Thus, digital technology may be seen as a modifiable factor in the relationship between cognitive function and HRQoL. We also found that with the continuous development and improvement of digital technology, digital usage behavior can have a profound impact on the cognitive function of patients with chronic diseases [18,19,20]. In addition to this, it is important that digital usage behavior, such as Internet use, can also have a certain impact on HRQoL [21]. Previous research in other populations found that using the Internet resulted in significantly higher PCS and physical pain scores [22]. According to self-determination theory, we believe that when individuals use the internet for leisure or work, their needs are met, which in turn produces positive long-term psychological outcomes such as quality of life [23]. So far, existing research is leaning towards the belief that an increase in digital usage behavior can improve the health of individuals [24]. However, there are also studies holding the opposite view; they believe that the long-term digital usage behavior reduces patients’ time for outdoor activities, and that a large amount of negative information received due to digital usage behavior is more likely to cause mood swings, which may have side effects on patients’ recovery [25]. Therefore, it is easy to find that digital usage behavior is likely to have various effects, so we propose another hypothesis: in middle-aged and older Chinese patients with diabetes, the relationship between cognitive function and HRQoL may differ considering digital usage behavior.

To date, most studies on the association of cognitive function with HRQoL have used cross-sectional designs, and have failed to elucidate the direction of influence between cognitive function and HRQoL due to the limitations of cross-sectional studies in terms of making causal inferences and explaining the direction of association. Therefore, it is necessary to further explore, especially for diabetic patients, the longitudinal association between the cognitive function and HRQoL, and to clarify the direction of their effects. As a longitudinal study, this study attempts to use a cross-lag model to explore the relationship between cognitive function and HRQoL in Chinese middle-aged and older diabetic patients, as well as the direction of this relationship and whether there were differences in digital usage behavior. This research is attempting to explore the longitudinal relationship between cognition and HRQoL in middle-aged and older diabetic patients, and to provide evidence which can promote their health.

It should be noted that, due to the existence of multiple databases in China, in addition to the China Health and Retirement Longitudinal Survey (CHARLS), most scholars use databases such as the China Household Finance Survey when researching topics related to public health [26]. There is more content regarding the physical and mental health of middle-aged and elderly people, so we chose CHARLS as our data source.

## 2. Materials and Methods

### 2.1. Participants

The China Health and Retirement Longitudinal Survey (CHARLS) is a representative follow-up survey of middle-aged and older people in China, chaired by the National Research Institute of Development at Peking University. The study protocol was approved by the Institutional Review Board of Peking University (approval number: IRB00001052-11015), and the study protocol complies with the ethical guidelines of the 1975 Declaration of Helsinki. In order to obtain the relevant data, we applied for the CHARLS database online on 31 August 2022, and approval was quickly obtained. This study used two waves of CHARLS data, from 2015 (T1) and 2018 (T2). CHARLS data can be accessed through its official website (https://charls.pku.edu.cn (accessed on 31 August 2022)).

Figure 1 shows the detailed process for including and excluding study participants. Wave 3 was used as the baseline data set for this study, and individuals who lacked information on HRQoL, cognitive function, digital usage behavior, and covariates were excluded from this study. Subsequently, new participants in 2018 were further excluded. Individuals lacking information on digital usage behavior, cognitive function, and covariates were also excluded from this study. In addition, in the sample selection, the research team excluded individuals with other diseases (such as cognitive diseases that may affect the detection process, mental diseases, and other diseases related to aging), and included individuals with diabetes only. In addition, the abbreviations of the main variables involved in the study are organized in Table 1.

### 2.2. Measurements

#### 2.2.1. Cognitive Functioning

This study obtained data on cognitive function from baseline data in 2015 and follow-up data in 2018. The baseline data stipulated by CHARLS is from 2011, but we include CHARLS 2015 (T1) as the baseline data in our study. CHARLS is similar to the cognitive assessment used in the American Health and Retirement Study, and has constructed cognitive function evaluation criteria from the same two aspects of memory and mental state [27,28]. Furthermore, previous studies have also used the CHARLS cognitive function assessment criteria for class-correspondence studies. Firstly, memory evaluation includes immediate word recall (0–10 points) and delayed word recall (0–10 points). Mental state is measured from three dimensions: orientation, visual construction, and mathematical performance. Orientation (0–5 points) is measured by asking respondents to name the date, day of the week, and season; visual construction is assessed by drawing a previously displayed picture (0–1 points); and mathematical performance (0–5 points) is measured by asking respondents to subtract 7 consecutive times from 100. The scores of the participants’ cognitive function are equal to the sum of the scores of memory and mental state. Cognitive function scores range from 0 to 31 points, higher cognitive function scores indicating better cognitive function. Finally, in order to obtain a composite measure of cognitive function, this study normalized and averaged the total cognitive function score by adding up memory and mental state scores [29].

#### 2.2.2. HRQoL

This study used a new scale construction based on the variables of the Short Form 36 (SF-36) and the CHARLS questionnaire to measure HRQoL in diabetic patients. The construction of the new scale was derived from the eight dimensions of SF-36, and the corresponding variables of CHARLS were selected to assess the following eight dimensions (Table 2): physical function (PF), role–body (RP), body pain (BP), general health (GH), vitality (VT), social functioning (SF), role–emotion (RE), and mental health (MH). The scores for the above eight dimensions were calculated by adding up the category scores and then converting the raw scores to a 0 to 100 scale. Scores from the eight subscales were aggregated into two overall scores according to the conceptual model of the SF-36 [30]. Physical function, body roles, body pain, and general perceptions of health were calculated as PCS, and mental health, vitality, emotional role, and social functioning were calculated as MCS. Although the HRQoL questionnaire based on CHARLS is slightly different from other HRQoL questionnaires, they all have similar focuses, including physical, emotional, and social diversity. The questionnaire has been determined to be effective in the Chinese population [31], and has already been used in related research [32].

#### 2.2.3. Socio-Demographic Variables

In order to minimize the possibility of other variables influencing the cognitive function–HRQoL relationship study, and to simplify the model, this research controlled for several specific covariates associated with cognitive function and HRQoL. According to previous studies, all covariates were based on baseline data (CHARLS 2015) [32]. Firstly, population control variables include age, gender, and education status. Secondly, since HRQoL can be divided into two parts, PCS and MCS, this study used different control variables in the models for cognitive function, PCS, and MCS. This research included depression as well as current smoking and drinking habits as control variables for PCS scores. In this research, PCS scores are also considered with marital status, depression, physical activity, and current smoking and drinking habits as control variables.

### 2.3. Statistical Methods

IBM SPSS Statistics, version 23 (IBM SPSS Statistics, Armonk, NY, USA) and Mplus version 8.0 are used for data analysis. Descriptive analyses of diabetic patient characteristics, cognitive function, and PCM and MCS of HRQoL were performed. The relationship between cognitive function and HRQoL in diabetic patients at T1 and T2 was assessed using the Pearson correlation test. A repeated-measures analysis of variance was performed to assess the differences between these variables with and without the digital usage behavior. Furthermore, cross-lagged panel structural equation modeling (SEM) was used to assess the longitudinal association between cognition and HRQoL at T1 and T2. Finally, a multi-group test was performed to assess whether digital usage behavior was making a difference in this relationship. Finally, regarding the grouping of digital technologies as well, previous studies were referred to [24]. Digital technology usage was obtained from 2015 baseline data. In the CHARLS 2015 questionnaire, internet usage was measured using the following question whether respondents have used the internet in the last month (0 for no, 1 for yes).

## 3. Results

### 3.1. Descriptive Statistics

The variable descriptive statistics and correlation analysis results are shown in Table 3 and Table 4. The correlation analysis shows that there is a significant correlation between cognition at T1 and T2 and HRQoL levels at T1 and T2, indicating that cognitive function and HRQoL levels in middle-aged and older diabetic patients have a certain relationship, which is showing stability. Meanwhile, simultaneous and sequential correlations between cognitive function and HRQoL levels are also significant. The correlation coefficients between cognitive function and HRQoL at T1 are 0.28 (*p* < 0.01) and 0.37 (*p* < 0.01); the correlation coefficients between cognitive function at T2 and HRQoL at T2 are 0.32 (*p* < 0.01) and 0.30 (*p* < 0.01); the correlation coefficients between cognitive function at T1 and HRQoL at T2 are 0.25 (*p* < 0.01) and 0.22 (*p* < 0.01); and the correlation coefficients between HRQoL at T1 and cognitive function at T2 are 0.32 (*p* < 0.01) and 0.36 (*p* < 0.01). This indicates that cognitive function is basically consistent with the synchronous and stable correlations of HRQoL levels, which is in line with the basic assumptions of the cross-lag design.

### 3.2. Stability Analysis of Cognitive Function and HRQoL

With cognitive function as the dependent variable, a 2 (test time: T1/T2) × 2 (digital usage behavior: use/non-use) repeated measures square analysis was performed. The results show that the testing time is the main and most significant effect (F = 11.21, *p* < 0.001, η^2^ = 0.01). The cognitive function level at T2 is significantly lower than at T1, and there are developmental differences. The main effect of digital usage behavior is significant (F = 36.325, *p* < 0.001, η^2^ = 0.06), and the cognitive function of patients using digital technology is significantly better than those patients without digital technology. The interaction between the two is not significant (F = 2.22, *p* > 0.05, η^2^ = 0.04).

With PCS as the dependent variable, a 2 (test time: T1/T2) × 2 (digital usage behavior: use/non-use) repeated measures square analysis was performed. The results show that the testing time is the main and most significant effect (F =16.25, *p* < 0.001, η^2^ = 0.03), the PCS level of the post-test is significantly lower than that of the pre-test, and there is a developmental difference. The main effect of digital usage behavior is significant (F = 36.54, *p* < 0.001, η^2^ = 0.06), and the PCS of patients using digital technology is significantly better than those not using digital technology. The interaction between the two is not significant (F = 0.18, *p* > 0.05, η^2^ = 0.00).

With MCS as the dependent variable, a 2 (test time: T1/T2) × 2 (digital usage behavior: use/non-use) repeated measures square analysis was performed. The results showed that the testing time is the main and most significant effect (F = 44.93, *p* < 0.001, η^2^ = 0.07), the MCS level of the post-test is significantly lower than the pre-test, and there is a developmental difference. The main effect of digital usage behavior is significant (F = 50.97, *p* < 0.001, η^2^ = 0.08), and the MCS of patients using digital technology is significantly better than those not using digital technology. The interaction between the two is significant (F = 4.43, *p* < 0.05, η^2^ = 0.08).

### 3.3. Cross-Lagged Analysis of Cognitive Function and HRQoL

In the cross-lag model which this study applied, HRQoL at T2 was predicted by cognitive function at T1, and cognitive function at T2 was predicted by HRQoL at T1. After the corresponding control variables were added to the models, both models achieved acceptable fitness criteria (Model 1: RMSEA = 0.093; CFI = 0.994, TCL = 0.912; Model 2: RMSEA = 0.098; CFI = 0.984, TCL = 0.901).

The cross-lag relationship between cognitive function and PCS scores at two points in time is shown from the test of Model 1 in Figure 2. First, in the same time period, the baseline association between cognitive function and PCS is significantly positive (B = 0.19, *p* < 0.01), implying better cognitive performance in middle-aged and older diabetic patients with higher PCS scores at T1, and vice versa. Second, there is no significant correlation between cognitive function at T1 and PCS at T2 (B = 0.05, *p* > 0.01), but PCS at T1 is positively correlated with cognitive function at T2 (B = 0.12, *p* < 0.01). This suggests that there is a positive and significant relationship between PCS and cognitive function in middle-aged and older diabetic patients over time, but early cognitive function in middle-aged and older diabetic patients does not affect later PSC.

The cross-lag relationship between cognitive function and MCS scores at the two points in time is shown in Model 2 of Figure 2. First, in the same time period, the baseline association between cognitive function and MCS is significantly positively correlated (B = 0.31, *p* < 0.01), which implies a better cognitive performance in middle-aged and older diabetic patients with higher MCS scores at T1, and vice versa. Second, there is no significant correlation between cognitive function at T1 and MCS at T2 (B = 0.05, *p* > 0.01), but there is a significant positive correlation between MCS at T1 and cognitive function at T2 (B = 0.14, *p* < 0.01). This suggests that there is a positive and significant relationship between MCS and cognitive function in middle-aged and older diabetic patients over time, but early cognitive function in middle-aged and older diabetic patients does not affect later MCS. In conclusion, there may be a one-way causal relationship between cognitive function and HRQoL, and the causal direction is from HRQoL to cognition.

### 3.4. Heterogeneity Analysis

In order to investigate whether the cross-lag relationship between cognitive function and HRQoL differs in digital usage behavior, the study performed a multi-group analysis. All diabetic patients were divided into two groups, one representing non-digital usage behavior (Figure 3), and the other representing digital usage behavior (Figure 4). In the grouped cross-lag model, the corresponding control variables were also added to the model. The models in Figure 3 and Figure 4 met fitness criteria (Figure 3: Model 3: RMSEA = 0.073, CFI = 0.996, TCL = 0.901; Model 4: RMSEA = 0.100, CFI = 0.990, TCL = 0.936. Figure 4: Model 5: RMSEA = 0.073, CFI = 0.996, TCL = 0.941; Model 6: RMSEA = 0.079, CFI = 0.970, TCL = 0.936). In both sets of models, the study focused on the relationship between cognitive function and HRQoL at a point in time, as well as their relationship at the time of follow-up research.

For middle-aged and older diabetic patients using digital technology, although the model could be fitted, cognitive function at T1 did not significantly predict HRQoL at T2 (PCS: B = 0.30, *p* > 0.01; MCS: B = 0.14, *p* > 0.01), and vice versa (PCS: B = 0.06, *p* > 0.01; MCS: B = 0.05, *p* > 0.01) (see Figure 3). For diabetic patients who did not use digital technology, although T1 cognitive function did not significantly predict T2 HRQoL (PCS: B = 0.04, *p* > 0.01; MCS: B = 0.05, *p* > 0.01), T1 HRQoL significantly predicted T2 cognitive function (PCS: B = 0.10, *p* < 0.01; MCS: B = 0.11, *p* < 0.01), and the results were consistent across all diabetic patients (see Figure 4). This shows that the cross-lag model of cognitive function and HRQoL in middle-aged and older diabetic patients shows variance in the use of digital technology, that is, this relationship will be affected by digital usage behavior.

## 4. Discussion

The present study aimed to further deepen our understanding of the association between cognition and HRQoL. This study used a cross-lagged model and controlled for the corresponding covariates in order to verify the longitudinal relationship between cognition and HRQoL in middle-aged and older diabetic patients. The corresponding findings suggest that this relationship is unidirectional in the cross-lag model, that is, only HRQoL at T1 can predict cognitive function at T2, and cognitive function at T1 does not predict HRQoL at T2. This study further explored whether the use of digital technology would lead to different manifestations of this relationship, and divided the study population into two categories: those who used digital technology and those who did not. The results found that this one-way lag relationship still existed in middle-aged and older diabetic patients who did not use digital technology, but was not significant in patients who used digital technology.

First of all, the study found that there are certain developmental differences in the cognitive function, PCS, and MCS levels of middle-aged and older diabetic patients. There is an increasing trend over time which is consistent with previous research results [33,34]. From an age perspective, not only are changes in cognitive function closely related to age [17,35], but HRQoL also has a distinct age-specific trajectory [36]. Therefore, as patients with diabetes age, the risk of cognitive decline and reduced PCS and MCS levels increases [15,37]. Second, diabetes is considered a risk factor for developing cognitive impairments [38]. Although some studies suggest that it may not accelerate the cognitive deterioration process, there is a general consensus in the academic community that diabetes increases the risk of abnormal cognition by increasing the incidence of cognition-related diseases or affecting blood sugar levels [37,38]. Furthermore, previous cross-sectional and longitudinal studies have identified that chronic diseases such as diabetes, as well as BMI, can affect HRQoL in a negative way [39,40]. Thus, this trend may be more obvious in the diabetic population.

It was found that the cognitive function of T1 was highly correlated with the PCS and MCS at T1 and T2, and the cognitive function of T2 was highly correlated with the PCS and MCS at T1 and T2, which was in line with the basic assumption of a cross-lag design. Our results are consistent with previous studies on other populations [9]. In the subsequent cross-lag analysis, the results were found to be consistent with previous studies [12]. The cognitive function, PCS, and MCS levels of middle-aged and elderly diabetic patients have a certain degree of lateral stability between T1 and T2, and HRQoL can predict cognitive function. However, there is no research to explain its internal mechanism. Verghese, J. et al., and Daviglus, M.L. et al., suggest that interventions to improve physical and general mental health can prevent or delay the onset of dementia [41,42]. Since HRQoL scores from CHARLS in this study include evaluations of physical activity and general mental health, it is believed that physical activity may be a plausible mechanism by which HRQoL can predict changes in cognitive function, based on previous studies. The mediating role of physical activity in this relationship should be further investigated in the future. In general, there are few studies focusing on the impact of HRQoL on cognitive function. In the future, more in-depth research is needed to explore the internal mechanism of how HRQoL affects cognitive function [43,44].

Third, although cognitive function at T1 was found to be significantly associated with HRQoL at T2 (PCS and MCS) in the initial correlation analysis, the predictive role of cognitive function at T1 on HRQoL at T2 in the cross-lag model was not significant, suggesting that cognitive function at T1 does not predict HRQoL at T2. This is consistent with previous studies [45]. One possible explanation is that the differences in cognitive function between the participants included are relatively small at baseline and at follow-up, as shown by a mean baseline cognitive function score of 15.73 and follow-up of 15.03. Therefore, there may not be enough individuals with the degree of cognitive function variation that would interfere with the study results. On the other hand, it may be that clinical health status alone does not determine a better quality of life [32], and for older adults, perceived life satisfaction appears to be more affected by ability to perform daily tasks [46]. Therefore, the influence of cognitive function on HRQoL may not be significant.

Finally, results show that there is a significant difference in the use of digital technology in the one-way predictive role of cognitive function and HRQoL, that is, the causal relationship between cognitive function and HRQoL is more applicable to diabetic patients who do not use digital technology. This finding is consistent with the study hypothesis. In terms of cognitive function, previous studies believe that the use of digital technology can slow the decline of cognitive function by increasing the cognitive reserve of individuals [18,47,48,49,50]. It has also been confirmed that in the diabetic population, digital usage behavior can improve the cognitive function of patients [51]; therefore, individuals who do not use digital technology tend to have greater changes in cognitive function, which are easier to detect than changes in individuals who do use digital technology. In terms of HRQoL, the use of digital technology has a direct positive impact on HRQoL [52], and for people with diabetes, digital usage behavior can have an indirect impact on HRQoL by stimulating positive lifestyle changes such as physical activity [53,54,55,56]. Therefore, it is believed that due to direct or indirect impact of digital usage behavior on cognitive function and HRQoL, digital usage behavior may have weakened the longitudinal association between cognitive function and HRQoL in middle-aged and older diabetic patients to some extent.

This study can help us to make some policy recommendations. First, the aging population has greatly increased the public health burden in China. Cognitive function and HRQoL in diabetic patients were affected to varying degrees. Therefore, it is necessary to improve the relevant medical service system for diabetics in China as soon as possible to ensure that the medical needs of the elderly are met. In addition, attention will need to be out into improving the Internet and medical service systems. In order to reduce the digital technology gap, promotion of internet use and healthy aging is encouraged.

This study did not focus on describing intra-individual variation due to the inherent limitations of the cross-lag model [57]. However, the cross-lag model used in this study still provides new evidence on the longitudinal relationship between cognitive function and HRQoL in middle-aged and older Chinese diabetic patients, and may provide new insights into the underlying mechanisms behind this relationship. Aside from the limitations of the model itself, there are several potential limitations to consider. First, there was a 3-year lag between baseline and follow-up, which may be considered too long to assess the cross-lag relationship between cognitive function and HRQoL. Future studies will need to test whether shorter and longer temporal associations differ in terms of gaining insight into the exact relationship between cognitive function and HRQoL in different conditions. Second, this study only selected the data of wave 3 and wave 4 of the CHARLS database, and did not include wave 1 or wave 2. This study used three questions that were only included after wave 3 in the evaluation criteria for the digital usage behavior. If there were enough data, it would obviously be better to include waves 1 and 2 in the analysis. In addition, a large portion of participants were excluded due to lack of necessary data. Although we controlled for demographic variables, this may still lead to selective bias in the data. Third, in the group analysis, due to the significant sample size difference between those who used digital technology and those who did not, results may have been inaccurate. However, the study results still show that the longitudinal relationship between cognitive function and HRQoL differs in terms of digital usage behavior. Finally, because patients with diabetes often have other chronic diseases of the elderly, future research is needed to further distinguish and identify the differences between diabetes and other chronic diseases of the elderly.

## 5. Conclusions

The cognitive function and HRQoL (PCS and MCS) in middle-aged and older Chinese diabetic patients have a certain degree of lateral stability, and both tend to increase over time. More importantly, this study found that HRQoL at T1 can significantly predict cognitive function at T2, but cognitive function at T1 cannot significantly predict HRQoL at T2. HRQoL may be an antecedent for middle-aged and older diabetic patients’ cognitive function, and this predictive relationship can be different depending on digital usage behavior. In the future, effective intervention on HRQoL of middle-aged and older diabetic patients can be considered to improve their cognitive function, thereby promoting the healthy development of middle-aged and older diabetic patients.

## Figures and Tables

**Figure 1 ijerph-19-12400-f001:**
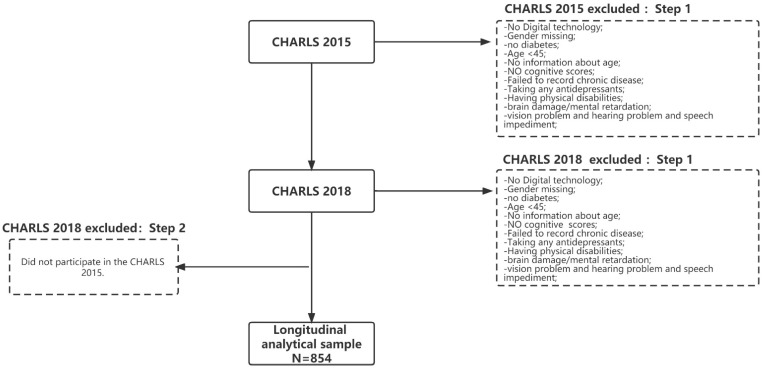
Sample Selection (*n* = 854). CHARLS, The China Health and Retirement Longitudinal Survey.

**Figure 2 ijerph-19-12400-f002:**
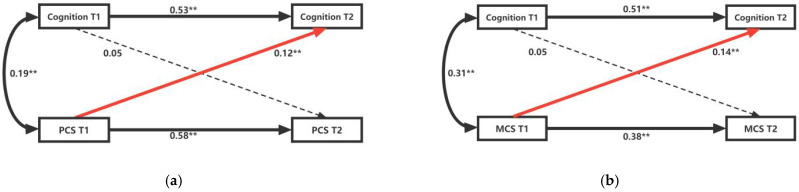
Cross-lag Model of Cognitive Function and HRQoL (all samples). (**a**) Model 1: the cross-lag relationship between cognitive function and PCS scores; (**b**) Model 2: the cross-lag relationship between cognitive function and MCS scores. PCS, physical component summary; MCS, mental component summary. ** denotes a statistically significant path coefficient *p* < 0.001.

**Figure 3 ijerph-19-12400-f003:**
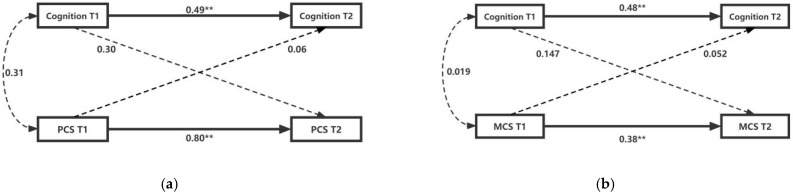
Cross-lag Model of Cognitive Function and HRQoL (digital usage behavior). (**a**) Model 3: the cross-lag relationship between cognitive function and PCS scores; (**b**) Model 4: the cross-lag relationship between cognitive function and MCS scores. PCS, physical component summary; MCS, mental component summary. ** denotes a statistically significant path coefficient *p* < 0.001.

**Figure 4 ijerph-19-12400-f004:**
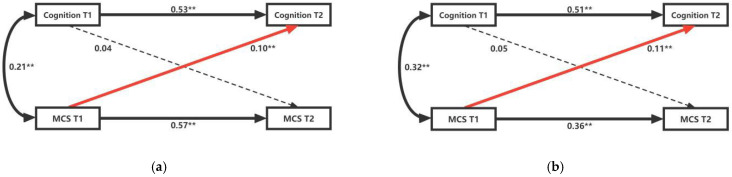
Cross-lag Model of Cognitive Function and HRQoL (no digital usage behavior). (**a**) Model 3: the cross-lag relationship between cognitive function and PCS scores; (**b**) Model 4: the cross-lag relationship between cognitive function and MCS scores. PCS, physical component summary; MCS, mental component summary. ** denotes a statistically significant path coefficient *p* < 0.001.

**Table 1 ijerph-19-12400-t001:** The meaning of abbreviations.

Variable	Abbreviations
The China Health and Retirement Longitudinal Survey	CHARLS
the Short Form 36	SF-36
health-related quality of life	HRQoL
physical component summary	PCS
mental component summary	MCS
physical function	PF
role-body	RP
body pain	BP
general health	GH
vitality	VT
social functioning	SF
role-emotion	RE
mental health	MH

**Table 2 ijerph-19-12400-t002:** Corresponding variables in the CHARLS data.

HRQoL (SF-36)	CHARLS Validity
PF	db001 db002 db003 db004 db005 db006 db007 db008 db009
RP	db016 db017 db018 db019 db020
BP	da041 da042s1 da042s2 da042s3 da042s4 da042s5 da042s6 da042s7 da042s8 da042s9 da042s10 da042s11 da042s12 da042s13 da042s14 da042s15
GH	da001 da002
VT	dc015 dc018
SF	da056s1 da056s2 da056s3 da056s4 da056s5 da056s6 da056s7 da056s8 da056s9 da056s10 da056s11 da056s12
RE	dc010 dc012
MH	dc009 dc011 dc014 dc016 dc017

CHARLS, The China Health and Retirement Longitudinal Survey. PF, physical function; RP, role–body; BP, body pain; GH, general health; VT, vitality; SF, social functioning; RE, role–emotion; MH, mental health.

**Table 3 ijerph-19-12400-t003:** Descriptive Statistics (*n* = 854).

Variables	All Participants	Digital Usage Behavior	
		Yes	No
Cognition (Mean ± SD)	15.73 ± 5.02	18.83 ± 4.09	14.62 ± 5.41
PCS (Mean ± SD)	71.82 ± 14.31	82.03 ± 8.97	70.71 ± 14.34
MCS (Mean ± SD)	56.48 ± 19.41	72.05 ± 11.92	66.30 ± 17.32
Sex (male, *n*, %)	467(54.71)	120 (70.03)	347 (50.87)
Age (Mean ± SD, years)	63.09 ± 8.73	59.61 ± 8.12	63.47 ± 8.72
Physical Activity (Mean ± SD)	1.44 ± 1.27	0.57 ± 0.91	1.54 ± 1.27
Depression (Mean ± SD)	18.70 ± 6.79	14.95 ± 0.44	19.10 ± 6.99
Marital status (Mean ± SD)	0.86 ± 0.34	0.93 ± 0.26	0.85 ± 0.35
Educational level (Mean ± SD)	2.41 ± 1.29	3.87 ± 0.97	2.26 ± 1.22
Smoking status (Mean ± SD)	0.22 ± 0.41	0.45 ± 0.50	0.19 ± 0.39
Drinking status (Mean ± SD)	0.39 ± 0.64	0.73 ± 0.67	0.35 ± 0.62

PCS, physical component summary; MCS, mental component summary.

**Table 4 ijerph-19-12400-t004:** Correlations between Cognitive Function and HRQoL (*n* = 854).

Variables	Cognition T1	Cognition T2	PCS T1	PCS T2	MCS T1	MCS T2
Cognition T1	1.00					
Cognition T2	0.60 **	1.00				
PCS T1	0.28 **	0.32 **	1.00			
PCS T2	0.25 **	0.32 **	0.63 **	1.00		
MCS T1	0.37 **	0.36 **	0.59 **	0.50 **	1.00	
MCS T2	0.22 **	0.30 **	0.44 **	0.55 **	0.48 **	1.00

** *p* < 0.001. PCS, physical component summary; MCS, mental component summary.

## Data Availability

The datasets presented in this study can be found in online repositories. The names of the repository/repositories and accession number(s) can be found below: https://charls.pku.edu.cn (accessed on 31 August 2022).

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
