# Peer review of "Longitudinal Relationship between Cognitive Function and Health-Related Quality of Life among Middle-Aged and Older Patients with Diabetes in China: Digital Usage Behavior Differences"

_ijerph, 2022, doi:10.3390/ijerph191912400_

Round 1
Reviewer 1 Report
The focus of this paper is on the relationship between cognitive function and health-related quality of life in middle-aged and older patients with diabetes, considering digital usage behavior differences. I have some concerns of this study even though it is an imporatant and interesting topic for readers.
1. Digital usage should be more specifically defined (ex. Using emails, application, SNS, devices, frequency.)
2. Please explain why the authors came up with the digital usage as an influencing factor of health-related quality of life. In the current form, there is a leap of logic in Introduction.
3. Guessing from the last paragraph of Introduction section, one of the aims of this study seems to be providing theoretical basis of such relationship. However, Discussion seemes to end up with introducing previous studies.
4. Please show the participants characteristics.
5. I suppose that there are few participants with cognitive impairment detected by using such tools. Did the authors adjust for participants' education level?
Author Response
请看附件

Reviewer 2 Report
Thank you for the opportunity to read this well written, clear and concise study. It was interesting and a pleasure to read.
The introduction is very complete and allows the reader to know the main topic of the research, informs about the purpose and importance of the work in the clinical field, and also answers the question posed in the scientific context. It includes previous works on the topic in question, explains the general problem of the research and specifies the objective of the study.
Materials and Methods
- - Please specify the type of study carried out in this work.
- - Every study must be registered, has it been so?
- - Please add a flow diagram of the participants.
- - Add a sample size calculation, referring to the literature and in sufficient detail to allow replication
The authors of this article have meticulously explained the results by providing relevant tables to what is explained in the text and have provided a clear and complete discussion by providing different works by other authors. Although, in the footer of each of the tables, you must add the meaning of the abbreviations that appear in each of them (also in the materials and methods section).
Lastly, the references section complies with the standards established by the journal and is homogeneous.
Reviewer 3 Report
Please see the attachment

Round 2
Reviewer 1 Report
The paper was revised along with reviewers' comments. Now I think it is suitable for the journal.
Author Response
Dear Reviewers,
Those comments are all valuable and very helpful for revising and improving our paper, as well as the important guiding significance to our research.
We would like to take this opportunity to thank you for all your time involved and this great opportunity for us to improve the manuscript.
Sincerely,
The Authors
Reviewer 2 Report
All the authors have correctly answered all the questions raised above, so it is accepted for publication.
Author Response

(The authors gave the same response as above.)

Reviewer 3 Report
please see the attachment.
